# Psychometric properties of ohip-edent b&h for conventional complete denture wearers

Srđan D. Poštić [1,2]*

**1** Department of Prosthodontics, University School of Dental Medicine, University of Belgrade, Beograd, Serbia, Europa, **2** Dental department, FZF- Faculty of Pharmacy and Medicine, University of Travnik, Travnik, Federation of Bosnia and Herzegovina

* srdjan.postic@stomf.bg.ac.rs

**Data Availability Statement:** Data are in ISRCTN https://doi.org/10.1186/ISRCTN11586190 which is visible at https://www.isrctn.com/ISRCTN11586190.

## Abstract

This study assessed psychometric properties of the Bosnian language version the Oral Health Impact Profile for Edentulous Patients (OHIP-EDENT), translated from the original English language version of and evaluated the Oral Health-related Quality of Life, in complete denture wearers before and after corrections of dentures. Specialist of dental prosthetic interviewed 117 edentulous patients before and after interventions. All patients had problems with their existing complete acrylic resin dentures. During the first visit, the patients were examined by prosthodontic specialists, who registered the status of the existing acrylic complete dentures and described interventions needed to improve denture quality. The patients were interviewed, and they completed the OHIP-EDENT questionnaire. Each patient was re-examined by a prosthodontic specialist one month after the new complete acrylic dentures had been delivered. This study's basic instrument was the Bosnian language version (B&H) OHIP-EDENT questionnaire. The questionnaire's internal consistency was first assessed by Cronbach alpha coefficient, which was 0.80, and after correcting of dentures 0.76. Significantly lower scores were found in domains of functional limitation ($p = 0.019$), psychological discomfort ($p = 0.010$), physical pain ($p = 0.003$), and handicap ($p = 0.041$) after old denture corrections, as well as significantly better quality of life (reduced OHIP-EDENT Summary scores; $p = 0.027$). The student's t test of the OHIP EDENT B&H general and group indexes regarding the patients' conditions after denture interventions showed significant reduction of the general index ($p = 0.02$) along with reductions of functional limitations ($p = 0.019$), pain ($p = 0.003$), physical disabilities ($p = 0.15$), psychological disability ($p = 0.002$), and handicap ($p = 0.002$). The OHIP-EDENT B&H exposed good psychometric properties.

## Introduction

According to World Health Organization (WHO) data, more than 40% of older people in Albania, Bosnia & Herzegovina, Bulgaria, Canada, Finland, Malaysia, and the UK suffer from total tooth loss [1]. One of the key factors contributing to the increasing number of patients with tooth loss is an increase in life expectancy [2]. Therefore, it is important for clinicians to

**Funding:** the author received no specific funding for this work.

**Competing interests:** No authors have copmeting interest.

know the oral health benefits that oral rehabilitations and different prosthetic constructions can offer patients. For this purpose, specific tools to measure oral health impact on quality of life have been developed in dentistry [3–6]. However, only a small number of studies have focused on measuring prosthetic therapy's degree of success or failure [7, 8].

The oral health related quality of life scales are tools for evaluating the effect of dental treatment on oral health and quality of life [9]. The Oral Health Impact Profile (OHIP) with 49 questions grouped under 7 subdomains provides a detailed analysis of oral health related quality of life based on the conceptual model of oral health [10]. A 14-item version of this scale (OHIP-14) [11], which covers the same 7 domains, demonstrated acceptable validity (this scale was developed to reduce the time required for answering questions and the number of incomplete answers). However, some parts of the OHIP-14 are inappropriate for complete dentures wearers [12]. Therefore, it was implemented the OHIP-EDENT scale, a 19-statement version, that can detect changes in oral health related quality of life before and after insertion of new complete dentures [12]. The OHIP-EDENT can be used to evaluate negative impacts specifically related to edentulous patients [13].

This study's aim was to assess the internal consistency of a translated version of the original OHIP-EDENT [14–16] and evaluate complete denture wearers' quality of life, with additional contributions from determined manual professional interventions on dentures, in terms of improvements.

## Materials and methods

This case-series study included 117 edentulous patients. Thirty-six (30.34%) were men, with an average age of 72.70 ± 9.04 years; 81 were women, with an average age of 73.10 ± 6.73 years.

All patients had problems with their existing complete acrylic resin dentures. For this reason, they consulted prosthodontic specialists at our dental hospital FZF- Faculty of Pharmacy and Medicine, Dental Department, University of Travnik, Federation of Bosnia and Herzegovina [17]. Initially this study was restricted only to the patients who had experience in therapy by complete denture of any kind (either conventional acrylic complete denture wearers or supradental complete denture wearers positioned on the remaining tooth roots or implants). This way all of an assessment of this study were restricted to the patients who previously wore an old pair of complete dentures and who need the complete reconstruction of occlusion by dentures again.

During the first visit, the patients were examined by prosthodontic specialists, who registered the status of the existing acrylic complete dentures and described interventions needed to improve denture quality. The patients were interviewed, and they completed the OHIP-E-DENT questionnaire. All of the patients' responses were recorded in the translated Bosnian version of the OHIP-EDENT questionnaire. The ID number was also assigned. Subsequently, a new set of complete dentures was made for, and delivered to, each patient.

Each patient was re-examined by a prosthodontic specialist one month after the new complete acrylic dentures had been delivered. At this time the new dentures were corrected for every patient, and interventions were recorded. The Bosnian language version of OHIP-E-DENT questionnaire, after corrections were completed again after the interventions; patients used the same ID numbers.

The original English version of the OHIP-19 was adapted through two translations carried out independently by two dentists with an advanced level of English [7, 18]. They discussed and produced a consensual Bosnian version of the OHIP-EDENT questionnaire. Thus, this study's basic instrument was the OHIP-EDENT questionnaire [14]. The Bosnian version of the OHIP-EDENT questionnaire was comprised of 19 questions related to the removable

denture wearers' quality of life. Answers were rated by the Likert scale with the following values: (1) = never; (2) = rarely; (3) = sometimes; (4) = almost always.

A specialist of prosthodontics completed following items:

- condition of acrylic teeth in the complete dentures;

- basal part of dentures;

- status of dentures;

- occlusion;

- functional equilibrium;

- retention;

- denture stability.

After obtaining ethic approval, the project 'Assessment of quality of life of acrylic removable denture wearers' was initiated in Central Bosnia [17]. This study assessed psychometric properties of the translated version of the original OHIP-EDENT questionnaire into the Bosnian language and evaluated quality of life of complete denture wearers before interventions. The first examination was before intervention, and the second examination was performed on the prostheses evaluated (denture corrections, relining, occlusal adjustments) after professional interventions.

Observing the quantitative–qualitative impact of the outcomes of corrections and interventions on dentures regarding quality of life, the researchers paid special attention to the parameters during the second investigation. The outcome was considered excellent if the condition of the teeth in the acrylic dentures did not require intervention due to satisfactory and balanced occlusion, as well as if the denture base was adapted with excellent stability and retention, with entire functional equilibrium achieved. Otherwise, the outcome was assessed as good.

The statistical analysis using the program IBM SPSS Statistics for Windows, Version 19.0 (Armonk, NY: IBM Corp. released 2011), was performed. Continuous variables are shown as mean ± standard deviation with range (minimum-maximum). Continuous variables before and after interventions was tested by Student's t test. Attributive variables are shown number with percent. Questionnaire reliability analysis was performed by determining Cronbach's (Cr) α. Correlations of questions were analyzed with the help of a correlation matrix (inter-item Pearson's correlations).

## Results

The questionnaire's internal consistency was calculated before (Crα = 0.80) and after the interventions (Crα = 0.76). The instrument's convergent validity was confirmed by a significant Pearson correlation between the OHIP-EDENT Summary Score and a prosthodontic specialist's assessment (0.49 to 0.59) (Table 1).

Summary scores before and after interventions and corrections (adjustments, relining) of dentures showed significant descending (Table 2).

Significantly lower scores were found in domains of functional limitation (p = 0.019), psychological discomfort (p = 0.010), physical pain (p = 0.003), and handicap (p = 0.041) after old denture corrections, as well as significantly better quality of life (reduced OHIP-EDENT Summary scores; p = 0.027).The frequency of respondents' responses (almost always, sometimes, never) is presented in (Fig 1).

The frequency of changed answers to individual items/questions is very important. Namely, during the second test, the patient's response rate 'never' increased by 3.1%; the frequency of

**Table 1. Correlation of ohip-edent-B&H summary score and all of domains of this score respecting the assessments of complete dentures before intervention by a specialist of prosthodontics.**

| Before intervention (denture corrections, relining, adjustments) | | OHIP-EDENT-B&H |
|---|---|---|
| OHIP-EDENT Summary score | | |
| | Sig. (two-tailed) | |
| | N | 117 |
| Assessment of denture acrylic teeth | Pearson correlation | 0.491** |
| | Sig. (two-tailed) | 0.000 |
| | N | 117 |
| Assessment of a denture in general | Pearson correlation | 0.548** |
| | Sig. (two-tailed) | 0.000 |
| | N | 117 |
| Assessment of denture's basal part adaptation | Pearson correlation | 0.593** |
| | Sig. (two-tailed) | 0.000 |
| | N | 117 |
| Assessment of denture occlusion | Pearson correlation | 0.509** |
| | Sig. (two-tailed) | 0.000 |
| | N | 117 |
| Assessment of Functional equilibrium | Pearson correlation | 0.493** |
| | Sig. (two-tailed) | 0.000 |
| | N | 117 |
| Assessment of Denture's Retention | Pearson correlation | 0.487** |
| | Sig. (two-tailed) | 0.000 |
| | N | 117 |
| Assessment of Denture's Stability | Pearson correlation | 0.491** |
| | Sig. (two-tailed) | 0.000 |
| | N | 117 |

** = statistically high significance

the patients who answered the questions with 'sometimes' increased by 10.2%; and the frequency of the number of patients who answered the questions with 'almost always' decreased by 13.3%. This shows that corrections and interventions helped improve the quality of life for complete denture wearers (Fig 2).

After the second examination, the 39.33% of the participants of this study have (qualifying their subjective felling having the dentures in their mouth) claimed that the state of their prosthesis is excellent, but 60.67% considered it good.

## Discussion

In the Bosnian-speaking area, as well as in the areas of ex-Yugoslav territories, various versions of the questionnaires to verify patients' oral-health quality of life have been used [19, 20].

Dental status and problems with prosthodontic rehabilitation affect social activities, such as work ability, family and parenting actions, and emotional life [21]. Prosthodontic rehabilitations with dentures improve the quality of life for patients of varying ages, but particularly for older patients. Dentures also increase patients' satisfaction with their orofacial system, which is evident through aesthetics, performance, and function [22, 23].

The delivery of acrylic dentures/complete acrylic dentures, overdentures, and implant-supported complete dentures in a patient's mouth inquires checking the ascertained quality of life, not only at the moment they begin therapy, but also at recalls. Therefore, it is particularly

**Table 2. The summary scores and scores of each domain ofthe OHIP-EDENT- B&H before and after corrections, adjustment and/or relining of complete dentures.**

|  |  | N | Mean | Standard deviation | Minimum | Maximum | Sig. (two-tailed)* |
|---|---|---|---|---|---|---|---|
| OHIP-EDENT | Before | 117 | 35.367 | 10.132 | 13.00 | 57.00 | 0.027 |
|  | After | 117 | 32.709 | 7.967 | 19.00 | 56.00 |  |
| Functional limitation | Before | 117 | 6.589 | 1.943 | 1.00 | 9.00 | 0.019 |
|  | After | 117 | 6.042 | 1.652 | 3.00 | 9.00 |  |
| Physical pain | Before | 117 | 7.863 | 2.392 | 2.00 | 12.00 | 0.003 |
|  | After | 117 | 7.401 | 2.042 | 4.00 | 12.00 |  |
| Psychological discomfort | Before | 117 | 3.803 | 1.475 | 2.00 | 6.00 | 0.010 |
|  | After | 117 | 3.418 | 1.219 | 2.00 | 6.00 |  |
| Physical disability | Before | 117 | 5.957 | 2.035 | 2.00 | 9.00 | 0.072 |
|  | After | 117 | 5.470 | 1.724 | 3.00 | 9.00 |  |
| Psychological disability | Before | 117 | 3.461 | 1.429 | 1.00 | 6.00 | 0.067 |
|  | After | 117 | 3.188 | 1.121 | 2.00 | 6.00 |  |
| Social disability | Before | 117 | 4.265 | 1.858 | 3.00 | 9.00 | 0.060 |
|  | After | 117 | 4.025 | 1.458 | 3.00 | 9.00 |  |
| Handicap | Before | 117 | 3.427 | 1.226 | 1.00 | 6.00 | 0.041 |
|  | After | 117 | 3.162 | 1.041 | 2.00 | 6.00 |  |

*T-test for equality of means

important to control the duration of prosthetic therapy success with removable acrylic dentures for a prolonged period of time, which requires good and permanent communication between the patient and therapist [24]. Also, control over the achieved therapy's effects is important after any significant corrections of occlusion of acrylic complete dentures, corrections of articulation, relining, rebasing, eventual loss of one or more denture supports regarding implants, etc. It is obligatory to consider how delivered conventional acrylic dentures fulfilled not only basic, but advanced, demands when positioned in an older patient's mouth,

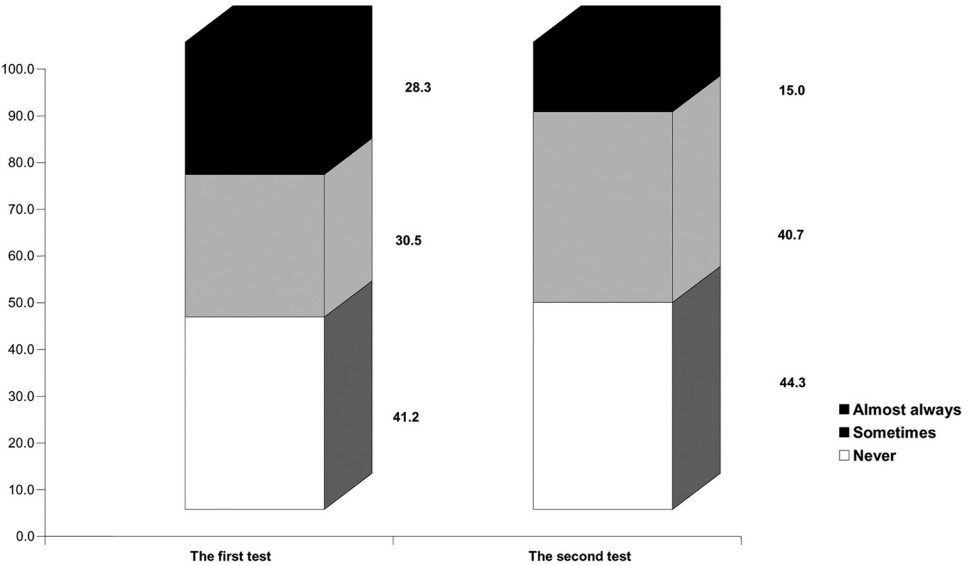

**Fig 1. The structure of the frequency of patients' answers to single items/questions of the ohip-edent-B&H questionnaire before and after interventions and corrections.**

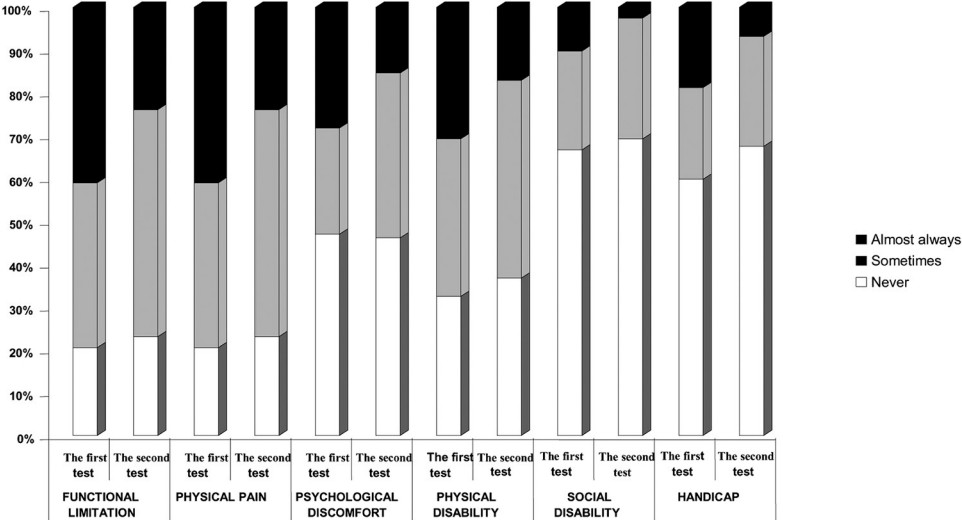

**Fig 2. The structure of frequencies of the answers (almost always, sometimes, never) to the single questions before and after corrections and interventions regarding group ohip-edent-B&H indexes.**

with no needs for relining or rebasing, as well as significant occlusal corrections during extended periods of time. However, despite all precautionary steps in the fabrication procedure, some objective processes, such as edentulous ridge reductions, compensations of lost hard and soft tissues after tooth extractions, adaptation period to occlusion established in dentures, or some transitional signs due to other (systemic or local) diseases during remissions, require interventions from prosthodontic specialists.

The effects achieved not only after initiation of prosthodontic treatment by acrylic complete dentures, but also after the corrections and interventions, will be reflected in the responses to the OHIP questionnaire. In this sense, patients experiencing improvements after the fabrication of new dentures, or after interventions and corrections, answered almost all of the questions with better grades. Good results and positive prosthetic effects can be considered if the patient answered questions during self-assessment of the prosthetic outcomes through the OHIP questionnaire mainly on items corresponding to chewing, pain/discomfort, presence of ulcers, dentures fit, denture retention, prosthetic comfort, and well-being with their mouths, which are the main complaints reported by edentulous patients seeking treatment (although answers showed how the prosthetic treatment could not resolve the food packing behind the prosthesis) [25].

In this study, the prosthetic outcomes concerning functional limitations (chewing, well-being with their mouths, etc.) improved, but the best results were obtained concerning elimination of psychological discomfort as well as improvements in terms of physical disability, social disability, and handicap.

In the municipality of the town of Travnik (Federation of Bosnia and Herzegovina), 42.62% of the population belongs to a risky age when it comes to the need for acrylic denture constructions. The total population aged 55 and over was 22.79%, and they receive basic health care from our health centers and dental department, where the current research was conducted [26]. For this reason, choosing an adequate instrument to examine the patients' oral health and quality of life is very important [23, 26, 27].

In order to understand the significance of the conducted research and its' positive implications on the selected population's oral health it should be noted that the National Health

Insurance Law in Art. 33 states that 'insured persons are entitled and have the rights to dental-prosthetic care and denture fabrication'. This assistance is free or provided with specific participation of the insured person [28].

The reliability of the instruments used in this study was principally measured in terms of Cronbach's alpha. The values of Cronbach alpha coefficients before interventions and after the interventions within the frameworks of this study were higher than the minimum of recommended value [18, 29]. Considering the Cronbach α values of 0.80 before interventions and 0.76 after the interventions in this research, it could be stated that the standard criteria for reliability is similar to comparable studies [30, 31]. This recommends OHIP-14 for use among the Bosnian elderly population.

The values of alpha Cronbach coefficients in this study were only slightly lower than some of coefficients that were listed in the studies of foreign authors who have also used OHIP index-14 (Short and reduced version of Oral health impact profile containing 14 questions), wherein the noted value was 0.89 for the population of examined edentulous patients in Jordan and Montenegro [32, 33], but 0.90 in Greece and Italy [34, 35], and 0.91 in Chile [36], and 0.97 in Poland [37] and 0.78 in Brazil [38].

Croatian authors translated the fifth question from OHIP-14 questionnaires differently and put it in a questionnaire [39]. In order to adjust the spirit of the language in the Japanese version, a further 14 questions were added to the fifth. The questionnaire's structure allows such changes since the overall score is not crucial for the index's validity [39].

## Limitations

This study is limited to conventional complete denture wearers only. Also, limitations of this study is the relatively small sample size. Further research should be directed towards performing this tested on a new significantly larger sample, as well as monitoring the application of this questionnaire in clinical practice.

## Conclusion

The OHIP-EDENT-B&H questionnaire can be considered as an effective instrument for measuring the outcome of treatment in the edentulous population S1 and S2 Figs.

## Supporting information

**S1 Fig. The structure of the frequency of patients' answers to single items/questions of the ohip-edent-B&H questionnaire before and after interventions and corrections.**
(TIF)

**S2 Fig. The structure of frequencies of the answers (almost always, sometimes, never) to the single questions before and after corrections and interventions regarding group ohip-edent-B&H indexes.** *T-test for equality of means.
(TIF)

**S1 Table. Correlation of ohip-edent-B&H summary score and all of domains of this score respecting the assessments of complete dentures before intervention by a specialist of prosthodontics.** ** = statistically high significance.
(DOCX)

**S2 Table. The summary scores and scores of each domain of the ohip-edent- B&H before and after corrections, adjustment and/or relining of complete dentures.**
(DOCX)

**S1 File. Original source with translation to domestic regional language.** Original source; Table 1A. (OHIP-EDENT-B and H)-domestic version of the influence of the specific components to oral status and oral health of edentulous patients which participated in the current study of SD Postic; additional questions are included in this table; Table 1B–Translations of questions in Table 1 from domestic language to English language and extension added for the purpose of the current study of SD Poštić.
(PDF)

## Acknowledgments

The author wishes to thank to Mrs. Olivera Stojanović and Associate Research Professor Nemanja Rančić for their help in statistical processing and calculations of the data of this study.

## Author Contributions

**Conceptualization:** Srđan D. Poštić.

**Data curation:** Srđan D. Poštić.

**Formal analysis:** Srđan D. Poštić.

**Investigation:** Srđan D. Poštić.

**Methodology:** Srđan D. Poštić.

**Project administration:** Srđan D. Poštić.

**Resources:** Srđan D. Poštić.

**Supervision:** Srđan D. Poštić.

**Validation:** Srđan D. Poštić.

**Visualization:** Srđan D. Poštić.

**Writing – original draft:** Srđan D. Poštić.

**Writing – review & editing:** Srđan D. Poštić.

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
