## [Decision Letter · Decision Letter 0]

17 Oct 2022

PONE-D-22-19472Psychometric properties of OHIP-EDENT B&H " for conventional complete denture wearersPLOS ONE

Dear Dr. Poštić,

Thank you for submitting your manuscript to PLOS ONE. After careful consideration, we feel that it has merit but does not fully meet PLOS ONE’s publication criteria as it currently stands. Therefore, we invite you to submit a revised version of the manuscript that addresses the points raised during the review process.

We look forward to receiving your revised manuscript.

Kind regards,

Walid Kamal Abdelbasset, Ph.D.

Academic Editor

PLOS ONE

Journal Requirements:

"No authors have copmeting interest"

 This information should be included in your cover letter; we will change the online submission form on your behalf

"No authors have copmeting interest."

Reviewers' comments:

Reviewer's Responses to Questions

**Comments to the Author**

1. Is the manuscript technically sound, and do the data support the conclusions?

Reviewer #1: Partly

Reviewer #2: Yes

2. Has the statistical analysis been performed appropriately and rigorously? 

Reviewer #1: Yes

Reviewer #2: Yes

3. Have the authors made all data underlying the findings in their manuscript fully available?

Reviewer #1: No

Reviewer #2: Yes

4. Is the manuscript presented in an intelligible fashion and written in standard English?

Reviewer #1: Yes

Reviewer #2: No

5. Review Comments to the Author

Reviewer #1: Introduction

I understand that being direct in the proposal is good for the manuscript, however, it lacks a better foundation! I suggest that the Introduction be revised and expanded.

Methods

OK, but the description is a little confusing. I suggest a review of the English for better suitability.

Results

4th paragraph: 39,33% indicated the prostheses as excellent and 78.67% as good. Added together this exceeds 100%!!! Please check!

Limitations

Wouldn't this be important information in the Methods?

Discussion

Well conducted.

Conclusion

OK

TAbles, Figures (graphs) and Supportin Information

Ok

Reviewer #2: Kindly note the following:

ABSTRACT: The methods section is extremely deficient and needs more information.

INTRODUCTION: This section is very short and does not give sufficient background about the topic under study.

Additionally, a thorough grammatical and spelling check-up throughout the whole manuscript is requested.

6. PLOS authors have the option to publish the peer review history of their article (what does this mean?). If published, this will include your full peer review and any attached files.

Reviewer #1: No

Reviewer #2: No

---

## [Author Response · Author response to Decision Letter 0]

24 Nov 2022

Comments to the Author

1. Is the manuscript technically sound, and do the data support the conclusions?

Reviewer #1: Partly

Reviewer #2: Yes

Author response: Thank you very much for your comments, the paper is now prepared according to the technical requirements of the journal.

2. Has the statistical analysis been performed appropriately and rigorously?

Reviewer #1: Yes

Reviewer #2: Yes

Author response: Thank you very much for your comments.

3. Have the authors made all data underlying the findings in their manuscript fully available?

Reviewer #1: No

Reviewer #2: Yes

Author response: Thank you very much for your comments.

4. Is the manuscript presented in an intelligible fashion and written in standard English?

Reviewer #1: Yes

Reviewer #2: No

Author response: Thank you very much for your comments, now the paper was reviewed by a native English teacher.

 

5. Review Comments to the Author

Reviewer #1: Introduction

I understand that being direct in the proposal is good for the manuscript, however, it lacks a better foundation! I suggest that the Introduction be revised and expanded.

Author response: Thank you very much for your comments. We have significantly expanded the introduction regarding the importance of these scales for assessing quality of life based on oral health assessment.

Methods

OK, but the description is a little confusing. I suggest a review of the English for better suitability.

Author response: Thank you very much for your comments, now the paper was reviewed by a native English teacher.

Results

4th paragraph: 39,33% indicated the prostheses as excellent and 78.67% as good. Added together this exceeds 100%!!! Please check!

Author response: Thank you very much for your comments, we corrected the mistake, it was excellent with 39.33% and good with 60.67%. 

Limitations

Wouldn't this be important information in the Methods?

Author response: Thank you for your comments. I moved limitations to Methods section and I extended it.

Discussion

Well conducted.

Author response: Thank you very much for your comments.

Conclusion

Author response: Thank you very much for your comments.

OK

TAbles, Figures (graphs) and Supportin Information

Ok

Author response: Thank you very much for your comments.

Reviewer #2: Kindly note the following:

ABSTRACT: The methods section is extremely deficient and needs more information.

INTRODUCTION: This section is very short and does not give sufficient background about the topic under study.

Additionally, a thorough grammatical and spelling check-up throughout the whole manuscript is requested.

Author response: Thank you very much for your comments. We have significantly expanded the introduction regarding the importance of these scales for assessing quality of life based on oral health assessment. 

Also, we have significantly expanded the abstract. 

Author response: Thank you very much for your comments, now the paper was reviewed by a native English teacher

---

## [Decision Letter · Decision Letter 1]

21 Dec 2022

Psychometric properties of ohip-edent B&H for conventional complete denture wearers.

PONE-D-22-19472R1

Dear Dr. Poštić,

We’re pleased to inform you that your manuscript has been judged scientifically suitable for publication and will be formally accepted for publication once it meets all outstanding technical requirements.

Kind regards,

Walid Kamal Abdelbasset, Ph.D.

Academic Editor

PLOS ONE

Additional Editor Comments (optional):

Reviewers' comments:

Reviewer's Responses to Questions

**Comments to the Author**

1. If the authors have adequately addressed your comments raised in a previous round of review and you feel that this manuscript is now acceptable for publication, you may indicate that here to bypass the “Comments to the Author” section, enter your conflict of interest statement in the “Confidential to Editor” section, and submit your "Accept" recommendation.

Reviewer #1: All comments have been addressed

Reviewer #2: (No Response)

2. Is the manuscript technically sound, and do the data support the conclusions?

Reviewer #1: Yes

Reviewer #2: (No Response)

3. Has the statistical analysis been performed appropriately and rigorously? 

Reviewer #1: Yes

Reviewer #2: (No Response)

4. Have the authors made all data underlying the findings in their manuscript fully available?

Reviewer #1: Yes

Reviewer #2: (No Response)

5. Is the manuscript presented in an intelligible fashion and written in standard English?

Reviewer #1: Yes

Reviewer #2: (No Response)

6. Review Comments to the Author

Reviewer #1: The revision made the manuscript a better version.

Thanks for the answers and for the effort to improve the manuscript.

Reviewer #2: (No Response)

7. PLOS authors have the option to publish the peer review history of their article (what does this mean?). If published, this will include your full peer review and any attached files.

Reviewer #1: No

Reviewer #2: No

---

## [Editor Report · Acceptance letter]

6 Jan 2023

PONE-D-22-19472R1 

Psychometric properties of ohip-edent B&H for conventional complete denture wearers. 

Dear Dr. Poštić:

I'm pleased to inform you that your manuscript has been deemed suitable for publication in PLOS ONE. Congratulations! Your manuscript is now with our production department. 

Kind regards, 

on behalf of

Dr. Walid Kamal Abdelbasset 

Academic Editor

PLOS ONE